# GAT++: Adaptive Relation-Aware Graph Attention Networks

## Abstract

Leveraging high-order structural semantics in knowledge graphs (KGs) is critical for modeling complex user preferences in recommendations. However, during multi-hop propagation, semantic noise arising from heterogeneous relation distributions obscures meaningful preferences, making it challenging to learn robust user-item representations. To address this challenge, we propose GAT++, a novel graph convolutional network that integrates relation-aware attention mechanisms with contrastive denoising regularization to learn robust and expressive user-item representations. At its core, GAT++ introduces an adaptive attention module that captures multiple semantic relation spaces by projecting entities into relation-specific subspaces and learning distinct relation weight distributions. To further suppress noise from high-order message passing, we introduce a contrastive regularizer that leverages multi-relation subgraph variants to enforce consistency across augmented views. Moreover, we develop a personalized denoising encoder that dynamically refines user-item representations end-to-end, removing the need for external data generation modules. We evaluate GAT++ on extensive real-world datasets across music, literature, and food domains. GAT++ achieves up to 34.81% improvement in Recall@N over strong baselines, demonstrating its effectiveness and generalizability across diverse recommendation scenarios.

## 1 Introduction

Effectively modeling structured semantic relationships in heterogeneous graphs is critical for tasks such as recommendation Wang et al. (2020a); Liu et al. (2019); Gharibshah & Zhu (2021), knowledge discovery, and retrieval. Heterogeneous graphs encode complex interactions through diverse relation distributions. However, the diversity and interdependence of these relations pose challenges for learning transferable, interpretable, and task-aware representations. Existing methods rely on fixed or manually defined importance scores

Table 1: The performance comparison of GAT and GAT++ shows a statistically substantial improvement in GAT++ ($p < 0.01$), with "RI" indicating the average relative improvement.

| | Recall@20 | Recall@50 | RI |
|---|---|---|---|
| GAT Veličković et al. (2018) | 0.0153 | 0.0457 | |
| **GAT++ (Ours)** | **0.1100** | **0.1500** | **424.24%** |
| | AUC | F1 | RI |
| GAT Veličković et al. (2018) | 0.7080 | 0.6341 | |
| **GAT++ (Ours)** | **0.8086** | **0.7240** | **14.18%** |

to represent relation semantics, which limits their adaptability across datasets and tasks. We propose a unified framework that combines a flexible architectural design, a universal contrastive loss, and adaptable data generation to dynamically model the varying importance of high-order semantic patterns (e.g., meta-paths) in an end-to-end manner. In recommendation tasks, our approach improves the expressiveness and robustness of user–item representations. While our primary focus is on recommendation, the framework effectively generalizes to a wide range of graph-based learning applications beyond user–item modeling.

Knowledge-enhanced recommendation has gained increasing interest for leveraging structured semantics from knowledge graphs (KGs) to improve representation learning and recommendation accuracy. Recent methods incorporate graph neural networks (GNNs) and attention mechanisms to enable high-order propagation and user preferences reasoning over multi-hop KG paths. Despite notable performance gains, they still face key challenges. A key issue is the semantic heterogeneity inherent in KGs: entities often engage in diverse, overlapping relation distributions, which render

typical aggregation schemes insufficient. Many models treat relations equally, ignoring their varying informativeness across latent semantic spaces. This neglect of relation spaces leads to indiscriminate message propagation, causing semantic dilution and noise accumulation in higher-order neighborhoods.

Moreover, mainstream methods apply entity-centric attention, overlooking relational semantics. This negligence results in coarse or relation-agnostic message propagation that fails to capture heterogeneous relational dependencies. For instance, Graph Attention Networks (GAT) Veličković et al. (2018) compute attention over neighboring entities without accounting for relation distributions, an oversimplification limiting the use of complex multi-relational graphs. Although self-attention-based models aim to capture dependencies and suppress irrelevant nodes Zhou et al. (2018; 2019); Sun et al. (2019), they often conflate multiple relation distributions within a shared embedding space, leading to degraded recommendation performance. A key underlying issue is the oversimplified assumption that all entities and their neighbors lie in a shared latent space, an assumption that overlooks the relational heterogeneity intrinsic to real-world knowledge graphs.

Second, existing methods are susceptible to semantic noise during higher-order propagation, where weakly related or irrelevant entities dilute meaningful preference and compromise model robustness. In recommendation systems, user preferences are influenced by both local interactions (with direct neighbors) and global structures (capturing multi-hop semantics). Effectively modeling the dynamic interplay between user interests and knowledge semantics is challenging, as user preferences are often entangled with evolving, context-dependent, and domain-specific knowledge along relation paths. It is a challenge to selectively amplify relevant semantics while suppressing spurious signals, which demands both theoretical complexity and computational demands.

Third, data sparsity in real-world user-item interaction graphs makes it hard to predict user interests. The absence of potential edges often causes attention mechanisms to overfit to noisy, distorting learned user and item representations and degrade recommendation quality. Moreover, the lack of reliable supervision in implicit feedback settings hinders effective training of KG-based models, as noisy or incomplete interactions can bias learning and impair generalization. These challenges underscore the need for a unified framework that enables fine-grained semantic reasoning, robust representation learning, and scalable optimization to enable knowledge-enhanced recommendations.

Our key insight is that an entity's contribution should vary across relational contexts. As user preferences propagate through multiple hops in a KG, the influence of each entity on the target depends on a combination of relation distributions, which jointly determine its overall contribution score. Building on this observation, our GAT++ explicitly models relation-specific semantic spaces and adaptively suppresses noise. In contrast to conventional GAT-based models that compute attention weights solely based on neighboring entities, GAT++ captures heterogeneous relational semantics by projecting entities into relation-specific subspaces using

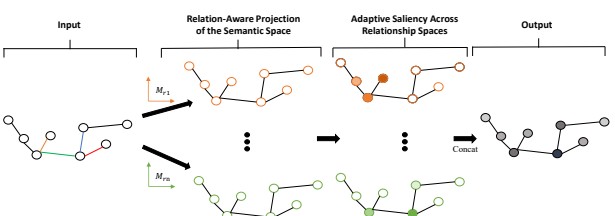

Figure 1: Overview of the adaptive attention mechanism in GAT++ for multi-relational representation learning: Given a graph with heterogeneous relations, node features are projected into relation-specific semantic spaces, where an adaptive saliency mechanism selects contextually relevant nodes.

learnable projection matrices. Relation attention distributions calculated by jointly considering head–tail entity interactions and their associated relational context, enabling fine-grained semantic discrimination across multiple relation spaces. As shown in Figure 1, GAT++ computes entity contribution scores across distinct relational semantic spaces, ensuring effective discrimination among relation distributions and reducing noise in high-order propagation. Empirical results on the Book-Crossing dataset (Table 1) demonstrate that GAT++ substantially outperforms GAT, with an average improvement of 424.24% in Recall@N across multiple recommendation metrics.

To reduce noise from irrelevant or weakly informative connections, GAT++ integrates a contrastive denoising regularization mechanism that leverages subgraph variants constructed from the most salient relation distributions. This regularizer enhances the robustness of learned embeddings by

maximizing agreement between semantically consistent views and filtering out noisy high-order signals through self-supervised subgraph alignment. Third, GAT++ introduces a personalized denoising encoder that dynamically generates user-item representations based on task relevance and jointly optimizes model parameters end-to-end. Unlike DR4SR+ Yin et al. (2024), which requires a three-stage training pipeline, our approach eliminates the need for external data generators or selectors, enabling automated dataset denoising and regeneration within a unified training process. Our contributions are as follows:

- We introduce GAT++, a graph attention network that models multiple semantic relation spaces through relation-specific attention and saliency-aware aggregation, improving robustness to high-order noise in knowledge graphs. To our knowledge, GAT++ is the first to introduce relation attention weight distributions.
- We propose a contrastive denoising regularization that addresses semantic noise from high-order propagation by generating subgraph variants from the most influential relation spaces to enhance representation consistency and suppress irrelevant paths.
- We propose a personalized denoising encoder that refines user-item representations end-to-end in a task-specific manner. This component enhances robustness and generalization under sparse supervision while eliminating the need for external data augmentation modules.
- We conduct extensive experiments on multiple public benchmarks, demonstrating that GAT++ consistently outperforms state-of-the-art recommendation models in both accuracy and robustness, especially under cold-start and sparse scenarios.

## 2 RELATED WORK

**User Interest Modeling** Traditional models like CF Sarwar et al. (2001) and FM Rendle (2012) have evolved into more expressive variants, such as NFM He & Chua (2017) and DFM Guo et al. (2017), which use deep neural networks (DNNs) to capture high-order feature interactions Sedhain et al. (2015); Chen et al. (2017); Wang et al. (2020a); Liu et al. (2019); Gharibshah & Zhu (2021); He et al. (2017); Rendle et al. (2020); Liang et al. (2018). Although these models enable deep feature crossing, they often introduce feature redundancy by indiscriminately modeling high-order interactions without accounting for contextual relevance. To address this, DIN Zhou et al. (2018) introduces an attention mechanism to focus on relevant user behaviors. DIEN Zhou et al. (2019) further extends this with an attention-enhanced GRU to model the temporal dynamics of user interests. However, methods that rely solely on user–item interactions exhibit limited generalization to broader application scenarios Togashi et al. (2021); Fan et al. (2019); Huang (2021). To enhance semantic context, knowledge graphs (KGs) have been integrated to enhance user preference modeling Wang et al. (2019c); Xian et al. (2019); Huang et al. (2021); He et al. (2020). However, propagating user preferences along multi-hop KG paths can introduce noise under complex semantic relationships, due to the lack of fine-grained modeling of relational semantics. Existing methods often fail to differentiate the varying importance of relations, weakening collaborative signals and underutilizing the rich structural semantics necessary for accurate preference modeling.

**Learning Fine-Grained Semantics** Incorporating knowledge graphs (KGs) into user and item representations has become central to recommendation research Zhang et al. (2016); Wang et al. (2018b); Xin et al. (2019); Tian et al. (2021). Recent methods use KG embeddings to capture semantic relationships Zhang et al. (2016); Wang et al. (2018b). For example, DKN Wang et al. (2018b) integrates knowledge into news content using TransD Ji et al. (2015), while DKFM Tian et al. (2023) applies TransE Bordes et al. (2013) to embed city-level information for destination recommendation. Other methods jointly learn entity and relation embeddings via propagation frameworks Wang et al. (2019c;b; 2018a); Xia et al. (2021); Hu et al. (2018); Wang et al. (2019a; 2020b). Additionally, contrastive learning Verma et al. (2021); Ruiter et al. (2019) has emerged as an effective way for enhancing semantic relationship embeddings Wu et al. (2021); Liu et al. (2021); Wei et al. (2022); Long et al. (2021). Existing methods emphasize collaborative signals but often overlook the complex semantic relationships in KGs, limiting their ability to capture fine-grained relational distinctions critical for accurate recommendations. Moreover, current attention mechanisms misalign with the KG's topology and semantic features, such as multi-hop paths and relation-specific contributions, hindering effective knowledge integration. To address these challenges, we propose

Figure 2: Overview of GAT++. User–item interactions are first encoded by a Personalized Denoise Encoder to obtain initial embeddings $e_u$ and $e_v$. These are refined through high-order propagation with Adaptive Relation-Aware Attention. Relation-specific subgraphs are used to generate diverse representations. The final embeddings are optimized via a multi-task objective, collaborative filtering loss $\mathcal{L}_{CF}$, knowledge graph loss $\mathcal{L}_{KG}$, and contrastive denoising loss $\mathcal{L}_{Noise}$.

an adaptive neural architecture that explicitly models the semantics of the KG, enabling accurate and interpretable recommendations.

## 3 PROBLEM DEFINITION

In recommendation systems, we define a user set $U = \{u_1, u_2, \ldots, u_M\}$ and an item set $V = \{v_1, v_2, \ldots, v_N\}$. Based on implicit feedback (e.g., clicks, views, purchases), we construct a user–item interaction matrix $Y \in \mathbb{R}^{M \times N}$, where $y_{uv} = 1$ indicates interaction and $y_{uv} = 0$ otherwise. Note that $y_{uv} = 0$ does not imply negative preference; it may simply indicate the item was unseen. To enrich user and item representations, we incorporate auxiliary knowledge from a knowledge graph $G = \{(h, r, t) \mid h, t \in \mathcal{E}, r \in \mathcal{R}\}$, where $h$ and $t$ are head and tail entities, and $r$ is a relation type. $\mathcal{E}$ and $\mathcal{R}$ denote the sets of entities and relations, respectively. To align items with semantic entities, we define an alignment set $A = \{(v, e) \mid v \in V, e \in \mathcal{E}\}$, where each pair $(v, e)$ links item $v$ to entity $e$. The objective is to predict the likelihood that a user $u$ will interact with an unseen item $v$, formalized as $\hat{y}_{uv} = F(u, v \mid \theta, G)$, where $F$ is the model, $\theta$ the learnable parameters, and $\hat{y}_{uv}$ the predicted interaction score.

## 4 METHODOLOGY

In this paper, we propose GAT++, a graph neural network that integrates a relation-aware attention mechanism, contrastive denoising regularization, and a personalized denoising encoder to learn robust and fine-grained user–item representations in multi-relational knowledge graphs (Figure 2).

### 4.1 ADAPTIVE RELATION-AWARE ATTENTION OF GAN++

The primary source of noise in knowledge graphs (KGs) arises from the complex semantics of relational spaces. To address this, GAT++ projects head and tail entities into relation-specific subspaces via a learnable projection matrix during attention computation. Our approach effectively represents entities' multi-faceted information and emphasizes distinct semantic dimensions. The influence of a tail entity $e^t$ on a head entity $e^h$ within the $r$ space is calculated as:

$$\text{Att}(\mathbf{h}, \mathbf{r}, \mathbf{t}) = \left[ \frac{(\mathbf{e}^h M_r) r^\top}{\sqrt{d}} \right] (\mathbf{e}^t M_r). \tag{1}$$

where $M_r$ is the projection matrix for relation $r$, and $d$ is the embedding dimension. GAT++ extracts features from all relation-specific subspaces in parallel and aggregates them via concatenation followed by a linear transformation:

$$S = \text{Concat}\left[\text{Att}(h, r_1, t), \ldots, \text{Att}(h, r_n, t)\right] W^H. \tag{2}$$

where $n$ is the number of relation spaces and $W^H$ is a learnable weight matrix. To normalize contributions and facilitate effective gradient flow, we apply a softmax function over the attention scores:

$$S' = \frac{\exp(S)}{\sum_{i=1}^{n} \exp(S)}. \tag{3}$$

This formulation enables GAT++ to prioritize semantically informative relations and entities while suppressing noise. During multi-hop propagation, user preferences are iteratively aggregated from neighbors across relation spaces:

$$e_o^{rn(l)} = S' e_o^{rn(l-1)}. \tag{4}$$

where $rn$ indexes the $n$ relation spaces, $o$ denotes a user or item, and $l$ is the propagation layer. This adaptive weighting mechanism enables GAT++ to effectively differentiate heterogeneous relational features, enhancing robustness and expressiveness in user preference modeling.

## 4.2 RELATION-AWARE DENOSING REGULARIZATION

To mitigate noise introduced during higher-order propagation in KGs, we incorporate a contrastive learning regularization that employs auxiliary self-supervised signals to denoise KG embeddings. By maximizing mutual information between augmented views, this mechanism preserves salient features of user and item representations and alleviates supervision sparsity. Existing approaches such as SGL Wu et al. (2021) and KGCL Yang et al. (2022) apply data augmentation to user–item interaction graphs, but they have notable limitations. SGL relies on random dropout, which may discard informative interactions critical for contrastive learning. KGCL, constrained to single-hop subgraphs, struggles to capture multi-hop relational semantics, limiting its ability to encode high-order user preferences and intent. In contrast, GAT++ utilizes subgraph variants across different relational spaces to construct contrastive views, improving the robustness and precision of graph-based collaborative filtering. At each propagation layer, GAT++ samples multiple subgraphs and selects the top two relation-specific variants to form augmented views of user–item interactions. This approach integrates external knowledge effectively by constructing two contrastive representations per node and applying a denoising regularizer that filters out noisy or irrelevant edges, eliminating the need for additional augmentation modules and reducing computational overhead. These subgraph-derived representations are concatenated to generate diverse user and item embeddings. Positive pairs are constructed from different relation views of the same node, while negative pairs are drawn from other nodes across the graph. Given two augmented representations of a user or item, $(e_{u1}, e_{i1})$ and $(e_{u2}, e_{i2})$, the contrastive loss is defined as:

$$L_{noize} = -\sum_{o \in G} \log \frac{\exp(s(e_{o1}, e_{o2})/\tau)}{\sum_{o' \in G, o' \neq o} \exp(s(e_{o1}, e_{o'2})/\tau)}. \tag{5}$$

where $\tau$ is a temperature hyperparameter and $s(\cdot)$ denotes cosine similarity. Minimizing $L_{\text{noize}}$ encourages consistency between positive pairs while discriminating against negative samples, guiding the model to learn clear, task-relevant user–item representations.

## 4.3 PERSONALIZED DENOISING ENCODER

To improve the robustness and task alignment of user and item embeddings, GAT++ incorporates a Transformer-based personalized denoising encoder that adaptively refines interaction sequences by capturing individualized behavior patterns and suppressing irrelevant signals. Unlike generative augmentation methods, this encoder functions as an attention-driven generator, selecting informative interactions while downweighting noisy or spurious inputs. Integrated into the end-to-end training pipeline, it enables joint optimization of input refinement and model learning, effectively addressing sparsity and noise by aligning learned representations with recommendation objectives. The encoder processes the raw interaction sequence of a user $u$ (or item $i$) as input tokens. Each token is constructed by concatenating the corresponding embedding $e_u$ (or $e_i$) with a timestamp embedding.

$$e_o = \text{Transformer}\big([e_{o,1}, \ldots, e_{o,T}]\big). \tag{6}$$

where $T$ denotes the sequence length and $o \in \{u, i\}$. The output of the [CLS] token, $e_o$, serves as the denoised representation and is subsequently used in GAT++ propagation. This module is trained jointly with the main model in an end-to-end manner. User–item interactions in the sequence are

Table 2: The Recall@K results in top-K recommendations are presented, with "RI" representing the average relative improvements of GAT++. The improvement of GAT++ is statistically significant, with a $p$-value $< 0.01$.

| Models | Last.FM | | | | Book-Crossing | | | | Dianping-Food | | | |
|---|---|---|---|---|---|---|---|---|---|---|---|---|
| | Recall@10 | Recall@20 | Recall@50 | RI | Recall@10 | Recall@20 | Recall@50 | RI | Recall@10 | Recall@20 | Recall@50 | RI |
| BPRMF Rendle et al. (2012) | 0.0723 | 0.1002 | 0.1832 | 195.39% | 0.0518 | 0.0589 | 0.0809 | 69.18% | 0.1397 | 0.1746 | 0.3072 | 86.26% |
| CKE Zhang et al. (2016) | 0.0721 | 0.0987 | 0.1823 | 197.60% | 0.0524 | 0.0597 | 0.0812 | 67.63% | 0.1396 | 0.1742 | 0.3068 | 86.53% |
| KGCN Wang et al. (2019b) | 0.1493 | 0.1988 | 0.2927 | 57.13% | 0.0519 | 0.0562 | 0.1093 | 55.18% | 0.1680 | 0.2265 | 0.3218 | 58.04% |
| KGNN-LS Wang et al. (2019a) | 0.1196 | 0.1696 | 0.2799 | 81.52% | 0.0432 | 0.0547 | 0.1174 | 62.05% | 0.1712 | 0.2401 | 0.3590 | 48.38% |
| KGAT Wang et al. (2019c) | 0.1647 | 0.2517 | 0.3567 | 32.09% | 0.0506 | 0.0649 | 0.1191 | 43.54% | 0.1786 | 0.2721 | 0.3879 | 36.87% |
| JNSKR Chen et al. (2020) | 0.1512 | 0.2311 | 0.3274 | 43.89% | 0.0472 | 0.0608 | 0.1103 | 53.96% | 0.1834 | 0.2807 | 0.4003 | 32.88% |
| CKAN Wang et al. (2020b) | 0.2106 | 0.2618 | 0.3699 | 18.30% | 0.0505 | 0.0688 | 0.1364 | 34.81% | 0.2080 | 0.3063 | 0.4489 | 19.05% |
| **GAT++ (Ours)** | **0.2430** | **0.2930** | **0.4720** | **–** | **0.0650** | **0.1100** | **0.1580** | **–** | **0.2550** | **0.3500** | **0.5400** | **–** |

weighted differently according to the objective of the current scenario. To enhance item exposure, a key objective in industrial applications, we can pass all item embeddings through the personalized denoising encoder.

### 4.4 User-Item Preference High-order Modeling

To enhance semantic representation quality in KGs, GAT++ refines entity and relation embeddings to better capture their structural dependencies. For each triple $(h, r, t)$, we adopt the scoring function from TransR Lin et al. (2015):

$$g(h, r, t) = \|e_h^r W_r + e_r - e_t^r W_r\|_2^2. \tag{7}$$

where $e_h^r$ and $e_t^r$ denote the head and tail entity embeddings projected into the relation-specific space via the transformation matrix $W_r$. The knowledge graph embedding loss is formulated as:

$$\mathcal{L}_{\text{KG}} = \sum_{(h,r,t,t') \in \mathcal{T}} -\ln \sigma \left( g(h, r, t') - g(h, r, t) \right). \tag{8}$$

where $\mathcal{T}$ denotes the set of training quadruples and $\sigma(\cdot)$ is the sigmoid activation.

For recommendation learning, we compute the likelihood of a user interacting with an item via an inner product:

$$\hat{y}_{ui} = h^T \left( e_u^{rn} \odot e_i^{rn} \right). \tag{9}$$

where $\odot$ denotes element-wise multiplication and $h \in \mathbb{R}^d$ is a trainable prediction vector. The final user embeddings $e_u^{rn}$ and item embeddings $e_i^{rn}$ are obtained via multi-layer aggregation over relation-specific spaces, originating from their base embeddings $e_u$ and $e_i$. We further define the training set $\mathcal{R} = \{(u, i, j) \mid (u, i) \in \mathcal{P}^+, (u, j) \in \mathcal{P}^-\}$, where $\mathcal{P}^+$ and $\mathcal{P}^-$ represent observed (positive) and sampled unobserved (negative) user–item interactions, respectively. The recommendation loss is represented as:

$$\mathcal{L}_{\text{CF}} = -\sum_{(u,i,j) \in \mathcal{R}} \ln \sigma \left( \hat{y}_{ui} - \hat{y}_{uj} \right). \tag{10}$$

We jointly optimize the recommendation, KG embedding, and denoising objectives using a unified multi-task loss:

$$\mathcal{L}_{\text{GAT++}}^{\text{Rec}} = \mathcal{L}_{\text{CF}} + \lambda_1 \mathcal{L}_{\text{KG}} + \lambda_2 \mathcal{L}_{\text{Noise}} + \lambda_3 \|\Theta\|_2^2. \tag{11}$$

where $\lambda_1$, $\lambda_2$, and $\lambda_3$ are hyperparameters controlling the contributions of each task and the $L_2$ regularization term, and $\Theta$ represents the model parameters. GAT++ is optimized using Adam Kingma (2014) due to its efficiency and low memory footprint in large-scale settings.

## 5 Experiments

We perform extensive experiments to evaluate the performance of our GAT++, focusing on the following research questions:

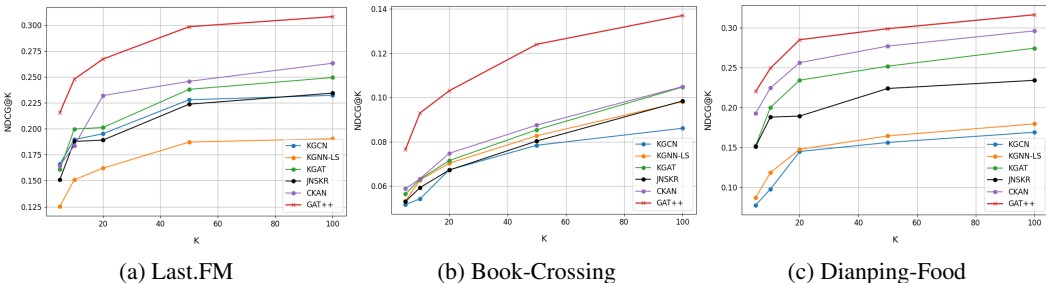

(a) Last.FM        (b) Book-Crossing        (c) Dianping-Food

Figure 3: Comparison of NDCG@K performance across three real-world datasets. GAT++ achieves statistically significant improvements over baselines, with $p < 0.01$.

Table 3: Performance comparison of GAT++ and leading methods on Last.FM, Book-Crossing, and Dianping-Food. The improvement of GAT++ is statistically significant where $p$-value $< 0.01$ level.

| Models | Last.FM | | Book-Crossing | | Dianping-Food | |
|---|---|---|---|---|---|---|
| | AUC | F1 | AUC | F1 | AUC | F1 |
| BPRMF Rendle et al. (2012) | 0.7596 | 0.6762 | 0.6591 | 0.6341 | 0.8326 | 0.7644 |
| CKE Zhang et al. (2016) | 0.7492 | 0.6743 | 0.6873 | 0.6254 | 0.8136 | 0.7416 |
| KGCN Wang et al. (2019b) | 0.8036 | 0.7098 | 0.6843 | 0.6321 | 0.8453 | 0.7753 |
| KGNN-LS Wang et al. (2019a) | 0.8098 | 0.7241 | 0.6782 | 0.6401 | 0.8521 | 0.7782 |
| KGAT Wang et al. (2019c) | 0.8329 | 0.7481 | 0.7352 | 0.6587 | 0.8462 | 0.7854 |
| JNSKR Chen et al. (2020) | 0.8014 | 0.7124 | 0.7332 | 0.6521 | 0.8701 | 0.7977 |
| CKAN Wang et al. (2020b) | 0.8441 | 0.7711 | 0.7471 | 0.6685 | 0.8778 | 0.8016 |
| **GAT++ (Ours)** | **0.8823** | **0.7984** | **0.8086** | **0.7240** | **0.8961** | **0.8180** |

- **RQ1**: How does GAT++ perform relative to other recommendation methods?
- **RQ2**: How effective is GAT++ in addressing the cold start problem?
- **RQ3**: What is the contribution of each key module in the GAT++ framework to its overall performance?

## 5.1 EXPERIMENTAL SETUP

### 5.1.1 DATASETS

GAT++ is evaluated on three widely used benchmark datasets: Last.FM, Book-Crossing, and Dianping-Food Wang et al. (2019a). For Last.FM and Book-Crossing, item-level knowledge is obtained by aligning items with entities in the Satori knowledge graph, whereas Dianping-Food leverages structured business metadata such as ratings, categories, and geographic locations. These datasets differ in domain, scale, and interaction sparsity, with detailed statistics summarized in Table 4. Specifi-

Table 4: Statistics of the three datasets.

| | **Last.FM** | **Book-Crossing** | **Dianping-Food** |
|---|---|---|---|
| #Users | 1,872 | 17,860 | 2,298,698 |
| #Items | 3,846 | 14,967 | 1,362 |
| #Interactions | 42,346 | 139,746 | 23,416,418 |
| #Entities | 9,366 | 77,903 | 28,115 |
| #Relations | 60 | 25 | 7 |
| #KG Triples | 15,518 | 151,500 | 160,519 |

cally, Last.FM comprises music listening records enriched with album and artist information; Book-Crossing provides explicit user ratings (ranging from 0 to 10) for books; and Dianping-Food contains over 10 million user—item interactions centered on restaurant reviews and ratings.

### 5.1.2 BASELINES

To assess the effectiveness of GAT++, we compare it with representative methods across four categories: collaborative filtering (CF), embedding-based models, non-sampling methods, and GNN-based frameworks. **BPRMF** Rendle et al. (2012) is a CF baseline using the Bayesian personalized ranking pairwise loss. **CKE** Zhang et al. (2016) jointly embeds structured, textual, and visual knowledge. **JNSKR** Chen et al. (2020) optimizes a non-sampling loss over all knowledge triples. GNN-based baselines include **KGCN** Wang et al. (2019b), which generates semantically enriched entity representations via high-order neighbor aggregation; **KGNN-LS** Wang et al. (2019a), which

Table 6: Performance comparison under cold-start setting.

| Models | Recall@20 | Recall@50 | NDCG@20 | NDCG@50 |
|--------|-----------|-----------|---------|---------|
| KGCN Wang et al. (2019b) | 0.0444 | 0.0996 | 0.0524 | 0.0697 |
| KGNN-LS Wang et al. (2019a) | 0.0463 | 0.0998 | 0.0543 | 0.0702 |
| KGAT Wang et al. (2019c) | 0.0499 | 0.1021 | 0.0562 | 0.0743 |
| JNSKR Chen et al. (2020) | 0.0403 | 0.0802 | 0.0347 | 0.0372 |
| CKAN Wang et al. (2020b) | 0.0514 | 0.1032 | 0.0578 | 0.0747 |
| **GAT++ (Ours)** | **0.0650** | **0.1200** | **0.0710** | **0.0940** |

introduces label smoothing regularization to improve robustness and generalization; **KGAT** Wang et al. (2019c), which introduces attention over multi-hop neighbors in a collaborative KG; and **CKAN** Wang et al. (2020b), which initializes entities with user-interacted items and similar users to explicitly encode collaborative signals.

### 5.1.3 IMPLEMENTATION DETAILS

We tune each baseline for optimal performance. The learning rate is searched over $\{10^{-3}, 2 \times 10^{-3}, 5 \times 10^{-3}, 10^{-2}, 2 \times 10^{-2}, 5 \times 10^{-2}\}$, embedding dimensions over $\{16, 32, 64\}$, and $L_2$ regularization over $\{10^{-5}, \ldots, 10^{-1}, 1\}$. Dropout rates are selected from $\{0.0, 0.1, \ldots, 0.9\}$, and task weights from $\{0.01, 0.02, \ldots, 0.9\}$. The number of GCN layers and optimal user/item subset sizes are dataset-specific: $\{3, 16, 64\}$ for Last.FM, $\{2, 32, 64\}$ for Book-Crossing, and $\{1, 16, 64\}$ for Dianping-Food. All datasets are split into training, validation, and test sets using a 6:2:2 ratio. We report AUC and F1 for CTR tasks, and RecallK and NDCGK for top-K recommendation.

### 5.2 OVERALL PERFORMANCE (RQ1)

We compare GAT++ with baseline models using results from Table 2, Table 3, and Figure 3, and highlight several key findings. First, GAT++ achieves a Recall@50 of 0.4720 on Last.FM, substantially outperforming the best baseline. On Book-Crossing, it reaches 0.1580, nearly double that of classical methods like BPRMF and CKE. On the complex Dianping-Food dataset, GAT++ attains the highest Re-

Table 5: Ablation study on the contributions of key components in GAT++ across three datasets.

| | Last.FM | | Book-Crossing | | Dianping-Food | |
|--------|---------|--------|---------------|--------|---------------|--------|
| | AUC | F1 | AUC | F1 | AUC | F1 |
| GAT++$_{w/o\ Att}$ | 0.8360 | 0.7620 | 0.7600 | 0.6880 | 0.8690 | 0.7920 |
| GAT++$_{w/o\ Noise}$ | 0.8640 | 0.7860 | 0.7830 | 0.7050 | 0.8820 | 0.8070 |
| GAT++$_{w/o\ Data}$ | 0.8705 | 0.7915 | 0.7720 | 0.6950 | 0.8880 | 0.8120 |
| **GAT++** | **0.8823** | **0.7984** | **0.8086** | **0.7240** | **0.8961** | **0.8180** |

call@50 of 0.5400, clearly surpassing other graph-based models. Second, GAT++ consistently outperforms the strongest baseline in both AUC and F1 across all datasets, with gains exceeding 8% on Book-Crossing. Third, it achieves the highest NDCG at all evaluated K, with strong improvements at lower K (e.g., K = 10, 20), where ranking precision is most critical. On Dianping-Food, GAT++ excels at top-K positions and maintains robust performance as K increases, demonstrating resilience to long-tail preferences. These consistent gains across metrics and datasets underscore GAT++'s effectiveness in modeling fine-grained user–item interactions over multi-relational, multi-hop knowledge graphs, resulting in relevant and semantically aligned recommendations.

### 5.3 ADDRESSING THE COLD START PROBLEM (RQ2)

The cold-start problem challenges recommender systems to make accurate predictions with limited interaction data. To assess model robustness under sparsity, we evaluate performance using only 20% of the training data on the Book-Crossing dataset (Table 4). GAT++ outperforms all baselines across four metrics, demonstrating its ability to learn meaningful representations despite data scarcity. Notably, it achieves a Recall@20 of 0.0650, 26.46% higher than the best baseline, and shows even greater gains in ranking metrics, with an NDCG@20 of 0.0710 and a 25.84% improvement in NDCG@50. These results highlight GAT++'s effectiveness in capturing fine-grained semantics and its resilience to the cold-start problem.

## 5.4 ABLATION STUDY (RQ3)

**Effect of Key Components** To assess the contribution of each GAT++ component, we conduct ablation studies using three variants: GAT++$_{w/o\,Att}$ excludes the adaptive relation-aware attention module, GAT++$_{w/o\,Noise}$ removes the contrastive learning component, and GAT++$_{w/o\,Data}$ omits task-specific regener-

Table 7: Comparison between GAT++ and DR4SR+.

| Model | Recall@10 | Recall@20 | Recall@50 | RI |
|---|---|---|---|---|
| DR4SR+ | 0.0462 | 0.0782 | 0.0906 | |
| GAT++ | **0.0650** | **0.1100** | **0.1580** | **61.24%** |

ated data from the personalized denoising encoder. As shown in Table 5, removing the attention module causes substantial performance drops on Last.FM, AUC falls from 0.8823 to 0.8360 and F1 from 0.7984 to 0.7620. Similar trends appear on Book-Crossing, where GAT++ outperforms GAT++$_{w/o\,Att}$ by 4.9 points in AUC and 3.6 in F1, highlighting the importance of relation-specific semantic modeling, especially in sparse settings. Eliminating contrastive learning consistently degrades performance across datasets, underscoring its role in filtering noisy or irrelevant interactions during representation learning. Excluding the personalized denoising encoder also leads to noticeable drops in AUC and F1, demonstrating its effectiveness in filtering noise and generating context-aware embeddings. Experimental results confirm that all three key modules in GAT++ contribute to performance gains, enabling robust and semantically enriched user–item representations.

**Effect of Dataset Regeneration** Table 7 compares GAT++ with the strong data generator DR4SR+ on the Book-Crossing dataset using Recall@K metrics. GAT++ consistently outperforms DR4SR+, with especially strong gains in short-list recommendations. It achieves a Recall@10 of 0.0650, surpassing DR4SR+'s 0.0462, a 40.7% improvement. This advantage extends to Recall@20 and Recall@50, where GAT++ scores 0.1100 and 0.1580 versus 0.0782 and 0.0906, yielding an overall relative

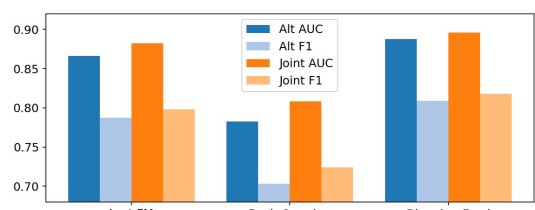

Figure 4: Performance comparison of learning strategies.

gain of 61.24%. Experimental results demonstrate GAT++'s effectiveness in capturing semantically relevant item relationships and improving retrieval across ranking depths. Unlike multi-stage methods like DR4SR+, GAT++ achieves superior performance in a unified, end-to-end framework.

**Effect of Learning Strategy** Figure 4 compares two multi-task learning strategies within the GAT++ framework: alternating optimization, which switches between contrastive and recommendation objectives by freezing one while training the other, and joint learning, which integrates both via a unified loss for simultaneous updates and tighter task coupling. Across all three datasets, joint learning consistently outperforms alternating optimization. On Last.FM, AUC improves from 0.8665 to 0.8823 and F1 from 0.7873 to 0.7984; on Book-Crossing, AUC rises from 0.7826 to 0.8086 and F1 from 0.7030 to 0.7240. GAT++ also leads on Dianping-Food. These gains show that joint learning enables more effective knowledge transfer between contrastive regularization and recommendation, promoting transferable user and item representations while reducing semantic noise during training.

## 6 CONCLUSION

This work addresses the challenges of high-order semantic noise due to the oversimplified relation modeling in heterogeneous graphs. We present GAT++, a unified graph attention network that captures fine-grained relational semantics by projecting entities into relation-specific subspaces and adaptively adjusting attention weights across multiple semantic spaces. To improve robustness, it introduces contrastive learning to enforce embedding consistency across subgraph views, effectively suppressing noise from high-order propagation. The personalized denoising encoder further refines user–item representations in a task-specific manner without external augmentation. Evaluated on extensive public benchmarks, GAT++ consistently outperforms state-of-the-art baselines in both sparse and cold-start settings. It offers a scalable and extensible solution for knowledge-aware recommendation and graph-based representation learning under semantic heterogeneity and sparse input data.

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
