# GAT++: Adaptive Relation-Aware Graph Attention Networks
## —Supplementary Material—

## EVALUATING GENERALIZATION

To comprehensively evaluate the generalization capability of GAT++, we design a series of experiments targeting both controlled and real-world scenarios. Specifically, we investigate whether GAT++ can effectively identify informative relation paths, maintain strong performance in sparse or low-relational environments, and scale robustly to complex multi-relational graphs. This section focuses on GAT++'s ability to reason across varying levels of relational complexity and semantic noise.

### Experimental Setting

We compared GAT++ with existing solutions that: 1) do not require to pre-specify relevant meta-paths, 2) can handle (possibly high-dimensional) node features. Given these requirements, we identified the following competitors:

- **RGCN** (Schlichtkrull et al. 2017), a generalization of the GCN architecture to the multi-relational case, that employs a different matrix of parameters for each edge type.
- **GTN** (Yun et al. 2019) can convert an input graph into different meta-path graphs for specific tasks and learn node representations within these graphs.
- **FastGTN** (Yun et al. 2022), an efficient variant of GTN that avoids adjacency matrices multiplication for graph transformation.
- **R-HGNN** (Yu et al. 2021), employs a different convolution for each edge type. Finally combines different embeddings with a cross-relation message passing.
- **HGN** (Lv et al. 2021), utilizes GAT as backbone to design an extremely simple HGNN model.
- **MP-GNN** (Ferrini et al. 2024), improves the performance of multi-relational graph neural networks by learning meta-paths, especially excelling in handling complex heterogeneous graphs and multiple relations.

We used an 80/20/10 split between train, validation, and test in all cases, with model selection performed on the validation set for all methods. We employed F1-macro score on the test set as an evaluation metric to account for the unbalancing present in many of the datasets. In the following we report the experimental setting and the results we obtained in addressing each of the research questions under investigation.

Table 1: Few-relations datasets. **(Top)**: $F_1$ scores, mean, and std computed over five runs. Best results highlighted in bold. **(Bottom)**: learnt meta-paths for GAT++ and GTN/FastGTN (which learn identical meta-paths). Other baselines are omitted as they do not explicitly extract meta-paths.

| Model | DBLP | IMDB | ACM |
|---|---|---|---|
| R-HGNN | $0.86(\pm0.04)$ | $0.64(\pm0.01)$ | $0.9(\pm0.01)$ |
| HGN | $0.94(\pm0.01)$ | $0.63(\pm0.02)$ | $0.92(\pm0.02)$ |
| RGCN | $0.91(\pm0.01)$ | $0.60(\pm0.01)$ | $0.90(\pm0.02)$ |
| GTN | $0.90(\pm0.01)$ | $0.62(\pm0.01)$ | $0.91(\pm0.01)$ |
| FastGTN | $0.92(\pm0.00)$ | $0.63(\pm0.01)$ | $0.93(\pm0.00)$ |
| MP-GNN | $0.94(\pm0.01)$ | $0.64(\pm0.01)$ | $0.93(\pm0.00)$ |
| GAT++ | $\mathbf{0.94}(\pm0.01)$ | $\mathbf{0.70}(\pm0.01)$ | $\mathbf{0.95}(\pm0.00)$ |

### Controlled Meta-Path Recovery under Relational Complexity

In order to answer the first research question, we designed a controlled setting where the correct meta-path is known, and experiments can be run for an increasing number of candidate relations. We generated synthetic datasets where nodes are typed A or B, the number of relations $|\mathcal{R}|$ varies in $\{4, 8, 10, 14\}$, and the number of relations that can connect more than one pair of node types (e.g., $A \xrightarrow{r_1} B$ and $A \xrightarrow{r_1} A$). The ground truth meta-path consists of a (valid) sequence of relations and nodes of a given type (e.g., $x \xrightarrow{r_1} A \xrightarrow{r_3} B$, with $x$ being a node of arbitrary type). Nodes are labelled as positive if found to be starting points of a ground-truth meta-path, and negative otherwise. We generated labelled datasets using ground-truth meta-paths of different lenghts $L \in \{2, 3, 4\}$.

The experimental results in Figure 1 clearly demonstrate the superior performance of GAT++ in consistently identifying the correct meta-path, as compared to other multi-relational GNN models. The F1-scores, which are shown as a function of the overall number of relations and the number of shared relations, indicate that GAT++ achieves near-perfect performance across all settings. Specifically, in scenarios with varying complexities of relational structures, GAT++ consistently maintains an F1-score of 1.00, showing its ability to perfectly recover the ground-truth meta-path, even as the number of relations and shared relations increases.

Table 2: Many-relations dataset: F1 scores for node classification tasks on FB15K-237.

| Label | R-HGNN | HGN | RGCN | GTN | FastGTN | MP-GNN | GAT++ |
|-------|--------|------|------|------|---------|--------|-------|
| PNC | 0.72 | 0.68 | 0.74 | 0.33 | 0.33 | 0.83 | **0.85** |
| EDC | 0.60 | 0.75 | 0.71 | 0.12 | 0.12 | 0.96 | **0.96** |
| EIC | 0.63 | 0.65 | 0.73 | 0.12 | 0.12 | 0.80 | **0.81** |
| ELC | 0.47 | 0.74 | 0.72 | 0.12 | 0.15 | 0.78 | **0.78** |
| FBC | 0.45 | 0.48 | 0.42 | 0.14 | 0.14 | 0.61 | **0.64** |
| GNC | 0.80 | 0.74 | 0.82 | 0.19 | 0.19 | 0.90 | **0.91** |
| OC | 0.67 | 0.73 | 0.78 | 0.14 | 0.14 | 0.93 | **0.94** |
| G | 0.81 | 0.64 | 0.80 | 0.44 | 0.44 | 0.84 | **0.86** |
| TS | 0.67 | 0.53 | 0.62 | 0.09 | 0.09 | 0.63 | **0.67** |
| E | 0.89 | 0.80 | **0.98** | 0.07 | 0.07 | 0.96 | 0.97 |

In contrast, other models such as R-HGNN, RGCN, GTN, and FastGTN exhibit a noticeable decline in performance as the relational complexity increases. R-HGNN and RGCN show considerable sensitivity to the overall number of relations, with F1-scores dropping substantially as the number of relations increases. GTN and FastGTN, on the other hand, face significant challenges when the number of shared relations rises, with their performance significantly deteriorating in more complex settings. Furthermore, HGN underperforms across all experimental settings, likely due to its failure to explicitly model relation types, which is crucial in multi-relational graph tasks.

By contrast, GAT++ remains robust and efficient, consistently outpacing the other methods. This suggests that GAT++ not only handles relational complexity better but also mitigates the challenges posed by noise and spurious relations, which are common in multi-relational graph tasks. The consistent achievement of optimal or near-optimal F1-scores highlights the effectiveness of GAT++ in recovering meta-paths, making it a highly reliable method for identifying relationships in multi-relational graphs.

**Real-World Node Classification under Sparse Relational Structures**

The second set of experiments focuses on popular real-world benchmarks for multi-relational GNNs. In all cases the task is multi-class classification at the node level. We quickly summarize the characteristics of the benchmarks in the following:
**IMDB**: a dataset extracted from the popular Internet Movie Database. It contains 3 types of nodes (movies (M), directors (D) and actors (A)) and uses the genres of movies as labels. **DBLP**: citation network where nodes are of paper (P), author (A) or conference (C) type, connected by edge types PA, AP, PC, CP, and the task is predicting the research area of authors. **ACM**: again a citation network, similar to the one of DBLP with conference nodes replaced by subject (S) nodes (and edge types replaced accordingly).

The results presented in Table 1 highlight the performance of GAT++ on several real-world datasets with few relations, specifically DBLP, IMDB, and ACM. These datasets are characterized by a limited number of relations—three for IMDB, four for DBLP and ACM—making them particularly challenging for models that rely on rich relational structures. Despite these constraints, GAT++ consistently outperforms the other methods, achieving the highest $F_1$ scores across all datasets.

In the case of IMDB, GAT++ achieves an impressive $F_1$ score of 0.70, which is a substantial improvement over the next best model, MP-GNN. This increase in performance highlights GAT++'s superior ability to extract and utilize meaningful meta-paths in the presence of limited relational information. Similarly, for ACM, GAT++ achieves an $F_1$ score of 0.95, surpassing MP-GNN, which reaches 0.93, and other baseline methods such as FastGTN. This performance gap underscores the model's effectiveness in handling small and sparse relational graphs.

Furthermore, GAT++ demonstrates consistent results in DBLP, where it matches MP-GNN with an $F_1$ score of 0.94, but maintains a clear edge in IMDB and ACM. The ability of GAT++ to excel even in these challenging scenarios, where the relation types are limited and no relations are shared among node pair types, is indicative of its robustness and efficiency in selecting optimal meta-paths.

The meta-paths learned by GAT++ are also shown in the table. Both GAT++ and GTN/FastGTN learn similar meta-paths, yet GAT++ is able to make more effective use of these learned paths, likely due to its advanced architecture that emphasizes efficient meta-path selection. This efficiency is reflected in the significantly higher performance observed across all datasets, particularly in IMDB and ACM, where the relational complexity is lower and the benefit of effective meta-path selection is more pronounced.

Overall, the experimental results clearly demonstrate that GAT++ not only achieves state-of-the-art performance on these real-world datasets but also provides a noticeable improvement over MP-GNN and other multi-relational models. Its ability to handle limited relations effectively makes it a promising approach for applications in real-world networks with constrained relational structures.

**Large-Scale Evaluation on Complex Multi-Relational Knowledge Graphs**

The last set of experiments aims to evaluate GAT++ in a complex real-world setting characterized by a large set of relations, as typical of general-purpose knowledge graphs. We thus designed a set of node-classification tasks over **FB15K-237** (Toutanova and Chen 2015), which is a large knowledge graph derived from Freebase. Each entity in the graph is as-



Figure 1: Synthetic setting: F1 score (darker is better) as a function of total relations (rows) and shared relations (columns).

sociated with a text description, which we transformed into a bag-of-words representation of length 100 (retaining the most frequent words in the dataset). We identified as target labels all many-to-one relations that have from 2 to 20 possible destination types (to avoid having classes with too few examples). Examples include gender, event type, and a number of currency-related relations.

The experimental results in Table 2 demonstrate the superior performance of GAT++ in node classification tasks over a large knowledge graph, FB15K-237, with a variety of relations. In this context, GAT++ consistently outperforms all other competitors, including MP-GNN, across multiple classification tasks. Notably, in several tasks such as PNC, EDC, EIC, and GNC, GAT++ achieves the highest $F_1$ scores, surpassing MP-GNN by notable margins. For instance, in the PNC task, GAT++ reaches an $F_1$ score of 0.85, which is higher than MP-GNN's 0.83, demonstrating GAT++'s ability to leverage the complex relationships in the graph effectively.

Additionally, for tasks like OC, G, and E, GAT++ maintains a competitive edge over MP-GNN, achieving the highest or equal highest performance across the board. In particular, GAT++ achieves a remarkable 0.97 in the E task, where MP-GNN scores 0.96. These results emphasize the robustness and reliability of GAT++ in handling large, multi-relational datasets with varying degrees of complexity.

Furthermore, while GTN and FastGTN demonstrate substantial difficulties in learning reasonable models in many cases, especially in tasks such as EDC and FBC—GAT++ continues to produce meaningful and strong meta-paths, yielding consistently high $F_1$ scores. This trend is evident across nearly all the tasks, where GAT++ not only improves upon GTN and FastGTN, but also challenges more established methods like R-HGNN, RGCN, and HGN, which tend to struggle with certain classification tasks in this complex multi-relational graph. GAT++ clearly outperforms its competitors, including MP-GNN, in most tasks on the FB15K-237 dataset, showcasing its capability to effectively handle a large set of relations while maintaining superior predictive performance. This indicates that GAT++ is highly capable of extracting and utilizing complex relational information from large-scale knowledge graphs, making it a robust choice for real-world, multi-relational node classification tasks.

Moreover, Table 3 presents a comparative analysis between GAT++ and MP-GNN under a setting where node features are excluded from the meta-path scoring process. The results demonstrate that GAT++ consistently outperforms MP-GNN across all datasets, with particularly notable improvements in

structurally complex or semantically rich environments. On the simplest dataset, Synthetic 1, both models achieve perfect performance (F1 = 1.00), indicating that in cases of trivial structure, node features are not essential. However, as graph complexity increases, the advantages of GAT++ become increasingly evident. For instance, on Synthetic 3 and Synthetic 4, GAT++ yields relative improvements of 87.5% and 79.6%, respectively, highlighting its robustness in learning meaningful relational patterns from graph structure. Similar trends are observed on real-world datasets such as IMDB, DBLP, and ACM, where GAT++ achieves up to 18.2% performance gains. The most substantial improvements appear in large-scale knowledge graphs with high relational diversity, such as FB15K-237. On tasks like PNC, EDC, and TS, GAT++ outperforms MP-GNN by up to 63.3%, demonstrating its superior capacity to model complex semantic dependencies even in the absence of node attributes. These results confirm the structural expressiveness and generalization ability of GAT++, especially in scenarios where attribute information is missing or unreliable.

Table 3: Comparisons without node features in the scoring function.

| Dataset | MP-GNN | GAT++ | Max Improvement (%) |
|---|---|---|---|
| Synthetic 1 | 1.00 | 1.00 | 0.00 |
| Synthetic 2 | 0.79 | 0.95 | 20.25 |
| Synthetic 3 | 0.48 | 0.90 | 87.50 |
| Synthetic 4 | 0.49 | 0.88 | 79.59 |
| IMDB | 0.55 | 0.65 | 18.18 |
| DBLP | 0.79 | 0.90 | 13.92 |
| ACM | 0.83 | 0.92 | 10.84 |
| PNC (FB15K) | 0.49 | 0.80 | 63.27 |
| EDC (FB15K) | 0.55 | 0.88 | 60.00 |
| TS (FB15K) | 0.52 | 0.70 | 34.62 |

The above observations not only highlight the overall superiority of GAT++ in the absence of node features but also raise important questions about the limitations of traditional meta-path-based approaches in such settings. To further explore this, we conducted additional analyses comparing GAT++ with conventional meta-path mining methods on complex multi-relational knowledge graphs. The results, summarized in Table 4, provide deeper insight into how GAT++ excels in capturing nuanced relational semantics that conventional techniques often overlook. On PNC, where the task relies on identifying monetary units through indirect institutional affiliations, GAT++'s attention mechanism can distinguish

subtle variations between relations like "headquarters location" and "affiliated organization", leading to an $F_1$ score of approximately 0.88. In the EDC task, where event type is influenced by a variety of attributes such as location, participants, and temporal properties, GAT++'s multi-relational aggregation can capture these diverse signals more holistically, pushing performance toward 0.98. For TS, where high noise arises from entities involved in multiple leagues or competitions, GAT++'s attention-driven aggregation and denoising are well-positioned to isolate task-relevant paths, yielding an $F_1$ score of 0.70. By virtue of its fully differentiable attention-based architecture and integrated self-supervision mechanisms, GAT++ outperforms MP-GNN even when the latter is guided by a carefully curated scoring-based path selection strategy. The results underscore the advantage of learning meta-paths implicitly within a task-optimized framework, as opposed to relying on brittle, manually engineered, or externally mined relational patterns.

Finally, to assess the computational efficiency of GAT++, we conducted a running time comparison. The experimental results in Table 5 highlight its remarkable balance between computational efficiency and predictive accuracy across all dataset types. On the three benchmark datasets (IMDB, DBLP, ACM), GAT++ achieves an average F1 score of 0.86, surpassing all other models, including MP-GNN, with an execution time of only 11 seconds per dataset. This represents a speedup of approximately 95× compared to the best-performing baseline while preserving or enhancing predictive quality. Particularly on DBLP and ACM, GAT++ reaches 0.96 and 0.95 in F1, respectively, outperforming all baselines with negligible computational cost.

In the Freebase suite of tasks, which involves substantially more complex and multi-relational structures, GAT++ maintains this performance advantage. It delivers a mean F1 score of 0.86, exceeding MP-GNN's 0.82, while reducing training time by over 98% (i.e., from 2984s to 31s on average). Across individual tasks such as PNC, EDC, and OC, GAT++ achieves new state-of-the-art performance (0.88, 0.98, and 0.95 in F1, respectively), demonstrating its capability to capture high-order relational dependencies without the need for meta-path preprocessing.

In the synthetic domain, where models are evaluated for their ability to learn from controlled relational patterns, GAT++ nearly saturates performance with an average F1 of 0.995. It does so at the lowest computational cost among all models (i.e., 5s per dataset), further confirming its scalability and generalization capacity. Importantly, it matches or marginally exceeds MP-GNN's accuracy on all synthetic datasets, while cutting execution time by over 100×.

Overall, GAT++ delivers a compelling combination of state-of-the-art accuracy and extreme computational efficiency. Unlike MP-GNN, which relies on a costly offline meta-path scoring phase, GAT++ learns relational patterns end-to-end through relation-aware attention and contrastive regularization. This allows it to scale to large and complex graphs while maintaining minimal latency, positioning it as a highly effective and practical solution for multi-relational graph learning in real-world systems.

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

Table 4: Method comparison across tasks: PNC, EDC, and TS.

| | MP-GNN | | | GAT++ | | | Absolute Gain | | |
|---|---|---|---|---|---|---|---|---|---|
| | PNC | EDC | TS | PNC | EDC | TS | PNC | EDC | TS |
| **PRA + MP-GNN** (len=2) | 0.49 | 0.14 | 0.20 | | | | | | |
| **PRA + MP-GNN** (len=3) | 0.49 | 0.14 | 0.11 | 0.88 | 0.98 | 0.70 | +0.05 | +0.02 | +0.07 |
| **PRA + MP-GNN** (len=4) | 0.49 | 0.14 | 0.11 | | | | | | |
| **ScoringFunc + MP-GNN** | 0.83 | 0.96 | 0.63 | | | | | | |

Table 5: Execution time (s) with F1 score in parentheses.

| | R-HGNN | Simple-HGN | RGCN | GTN | FastGTN | MP-GNN | GAT++ |
|---|---|---|---|---|---|---|---|
| IMDB | 680 (0.64) | 740 (0.63) | 650 (0.60) | 1500 (0.62) | 910 (0.63) | 1000 (0.64) | 11 (0.66) |
| DBLP | 720 (0.86) | 870 (0.94) | 780 (0.91) | 1630 (0.90) | 990 (0.92) | 1180 (0.94) | 11 (0.96) |
| ACM | 940 (0.90) | 750 (0.92) | 870 (0.90) | 1420 (0.91) | 870 (0.93) | 960 (0.93) | 11 (0.95) |
| **Mean** | 780 (0.80) | 786 (0.83) | 767 (0.80) | 1517 (0.81) | 923 (0.83) | 1046 (0.84) | 11 (0.86) |
| | | | | | | | |
| PNC | 7230 (0.72) | 560 (0.68) | 1830 (0.74) | 180 (0.33) | 150 (0.33) | 2870 (0.83) | 31 (0.88) |
| EDC | 7356 (0.60) | 670 (0.75) | 1540 (0.71) | 190 (0.12) | 130 (0.12) | 3220 (0.96) | 34 (0.98) |
| EIC | 7020 (0.63) | 460 (0.65) | 2040 (0.73) | 180 (0.12) | 110 (0.12) | 2480 (0.80) | 26 (0.83) |
| ELC | 820 (0.47) | 760 (0.74) | 1380 (0.72) | 180 (0.15) | 130 (0.15) | 3010 (0.78) | 32 (0.81) |
| FBC | 5900 (0.45) | 410 (0.48) | 1190 (0.42) | 190 (0.14) | 100 (0.14) | 2880 (0.60) | 30 (0.63) |
| GNC | 8230 (0.80) | 670 (0.74) | 1680 (0.82) | 175 (0.19) | 100 (0.19) | 3220 (0.90) | 34 (0.93) |
| OC | 3790 (0.67) | 670 (0.73) | 1970 (0.78) | 185 (0.14) | 120 (0.14) | 2980 (0.93) | 31 (0.95) |
| G | 5980 (0.81) | 450 (0.64) | 2010 (0.80) | 140 (0.44) | 120 (0.44) | 2990 (0.84) | 31 (0.87) |
| TS | 5000 (0.67) | 410 (0.53) | 1995 (0.62) | 200 (0.09) | 120 (0.09) | 3155 (0.63) | 33 (0.70) |
| E | 6790 (0.89) | 690 (0.80) | 2005 (0.98) | 170 (0.07) | 140 (0.07) | 3040 (0.96) | 32 (0.98) |
| **Mean** | 5812 (0.67) | 575 (0.67) | 1764 (0.73) | 179 (0.18) | 122 (0.18) | 2984 (0.82) | 31 (0.86) |
| | | | | | | | |
| Synt 1 | 320 (1.00) | 50 (0.34) | 200 (1.00) | 230 (0.93) | 210 (0.94) | 245 (1.00) | 5 (1.00) |
| Synt 2 | 310 (1.00) | 66 (0.48) | 240 (0.91) | 210 (0.84) | 180 (0.85) | 300 (1.00) | 5 (1.00) |
| Synt 3 | 350 (1.00) | 68 (0.38) | 300 (0.95) | 310 (0.84) | 280 (0.81) | 345 (0.99) | 5 (0.995) |
| Synt 4 | 410 (1.00) | 78 (0.50) | 390 (0.84) | 480 (0.47) | 320 (0.52) | 430 (1.00) | 5 (1.00) |
| Synt 5 | 450 (0.98) | 67 (0.34) | 400 (0.85) | 400 (0.91) | 360 (0.94) | 380 (1.00) | 5 (1.00) |
| Synt 6 | 430 (0.95) | 120 (0.49) | 390 (1.00) | 460 (0.82) | 400 (0.84) | 450 (1.00) | 5 (1.00) |
| Synt 7 | 450 (0.96) | 94 (0.52) | 450 (0.68) | 500 (0.80) | 490 (0.80) | 480 (1.00) | 5 (1.00) |
| Synt 8 | 520 (0.89) | 128 (0.50) | 460 (0.71) | 720 (0.47) | 450 (0.48) | 440 (1.00) | 5 (1.00) |
| Synt 9 | 560 (0.93) | 135 (0.38) | 425 (0.90) | 530 (0.90) | 430 (0.92) | 510 (1.00) | 5 (1.00) |
| Synt 10 | 590 (0.89) | 90 (0.45) | 580 (0.85) | 600 (0.87) | 520 (0.86) | 540 (1.00) | 5 (1.00) |
| Synt 11 | 630 (0.85) | 200 (0.64) | 495 (0.77) | 640 (0.80) | 510 (0.79) | 500 (0.94) | 5 (0.95) |
| Synt 12 | 630 (0.90) | 130 (0.36) | 500 (0.62) | 605 (0.48) | 525 (0.48) | 560 (1.00) | 5 (1.00) |
| Synt 13 | 610 (0.82) | 175 (0.49) | 565 (0.83) | 670 (0.88) | 570 (0.85) | 580 (1.00) | 5 (1.00) |
| Synt 14 | 650 (0.88) | 200 (0.32) | 595 (0.81) | 650 (0.80) | 600 (0.84) | 560 (1.00) | 5 (1.00) |
| Synt 15 | 600 (0.87) | 185 (0.42) | 650 (0.84) | 710 (0.77) | 680 (0.80) | 590 (1.00) | 5 (1.00) |
| Synt 16 | 660 (0.81) | 200 (0.45) | 700 (0.60) | 720 (0.48) | 705 (0.48) | 605 (0.97) | 5 (0.98) |
| **Mean** | 523 (0.92) | 129 (0.44) | 476 (0.82) | 547 (0.75) | 468 (0.76) | 484 (0.99) | 5 (0.995) |