# OpenReview forum: "GAT++: Adaptive Relation-Aware Graph Attention Networks"
_ICLR.cc/2026/Conference — Submitted to ICLR 2026_

### Official Review · Reviewer_FaRs · 2025-10-31

**Soundness:** 2
**Presentation:** 2
**Contribution:** 1
**Rating:** 2
**Confidence:** 4

**Summary:**

This paper propose a graph attention framework designed to improve knowledge-graph-based recommender systems by addressing high-order semantic noise. The method combines:
	1. Relation-aware multi-space attention, projecting entities into relation-specific subspaces,
	2. Contrastive denoising regularization, aligning subgraph variants,
	3. A personalized denoising encoder, a Transformer-style module that refines user-item representations.
Experiments on three public datasets (Last.FM, Book-Crossing, Dianping-Food) show large gains over baselines such as KGAT, CKAN, and DR4SR+.

**Strengths:**

1. The paper addresses an important problem, semantic noise and heterogeneity in multi-relational KGs for recommendation.
2. The overall architecture is coherent and modular, combining relational attention, contrastive regularization, and sequence-level denoising.
3. The empirical results are strong and consistent across multiple datasets and metrics (AUC, F1, Recall@K, NDCG).
4. The ablation studies are well-structured and demonstrate that each proposed component contributes positively.

**Weaknesses:**

1. **Limited Novelty.**
  The main contributions appear to be incremental combinations of existing approaches.
  Relation-specific projections or similar concepts have been already explored in *CompGCN*, *KGIN*, and *KGAT*.
  Contrastive denoising follows the paradigm of *SGL (SIGIR 2021)* and *KGCL (SIGIR 2022)*.
  and the personalized denoising encoder resembles *BERT4Rec* or *DIEN*.
  The claim of being the “first to introduce relation attention weight distributions” is not substantiated.

2. **Methodological Vagueness.**
  Core equations (1)–(4) lack precision.
  It is unclear how projection matrices \(M_r\) interact with entity embeddings (additive or multiplicative).
  The "adaptive saliency mechanism" is not formally defined, and the sampling of "subgraph variants from salient relations"
  lacks reproducible specification.
  The role and placement of the denoising encoder within the GAT layers remain ambiguous.

3. **Implausible Empirical Gains.**
  Reported improvements (e.g., +424% Recall@N over GAT) are unusually large for this domain.
  No variance, seed averaging, or significance testing details are provided despite repeated "p < 0.01" claims.
  Runtime and scalability analyses are missing even though the model adds computationally heavy components.

4. **Incomplete Baselines and Fairness.**
  The paper omits strong recent baselines such as *LightGCN*, *SimGCL*, *NCL*, and *KGCL*.
  *DR4SR+* is not a directly comparable baseline as it targets sequential recommendation rather than KG reasoning.
  Hyperparameter tuning fairness is not discussed.

5. **Clarity and Presentation Issues.**
  The text overuses vague terms such as "fine-grained semantics" and "semantic robustness" without clear definitions.
  Notation inconsistencies (e.g., \(L_{Noise}\) vs. \(L_{Noize}\), \(e_u^{R0}\) vs. \(e_u^{r_n}\)) hinder readability.
  Figures are schematic but fail to illustrate the precise data flow or architectural layering.

6. **Insufficient Analysis and Interpretation.**
  The paper lacks visualization or interpretation of learned attention weights, qualitative case studies, or
  error analyses that could clarify why the model performs better.

7. **No Theoretical Justification.**
  Claims such as "contrastive regularization maximizes mutual information" are not derived or proven.
  The discussion of "robustness to high-order noise" remains purely empirical and lacks formal analysis.

**Questions:**

1. How are the "relation-specific subspaces" and "salient relations" concretely defined and implemented?
2. How many contrastive views per node are sampled, and what is the computational cost of this procedure?
3. Were improvements verified over multiple random seeds?  Please include standard deviations in all tables.
4. How do you prevent potential data leakage between KG triples and user–item interactions?
5. Have you compared with modern contrastive recommendation baselines such as *SimGCL*, *NCL*, or *KGCL*?

---

> ### Author Response · Authors · 2025-11-18
>
> W1. We introduce GAT++, a graph attention framework that models multiple semantic relation spaces through relation-specific attention and saliency-aware aggregation, improving robustness to high-order noise in knowledge graphs. To our knowledge, GAT++ is the first to decouple attention weight distributions by relation. First, we propose a contrastive denoising regularizer that addresses semantic noise from high-order knowledge graph propagation by generating subgraph variants from the most influential relation spaces to enhance representation consistency and suppress irrelevant knowledge paths. Second, we propose a personalized denoising encoder that refines user-item interactions end-to-end in a task-specific manner. By eliminating the need for external data augmentation modules, this component enhances robustness and generalization under sparse supervision. Third, we conduct extensive experiments on multiple public benchmarks, where GAT++ consistently outperforms state-of-the-art recommendation models in accuracy and robustness across diverse datasets.
>
> W2, Q1 & Q2. In the paper, the relation projection matrices interact with entity embeddings multiplicatively. The adaptive saliency mechanism essentially performs relation space-level attention on a multi-relational KG. We construct and sample relation-subgraph views as follows: at each propagation layer, relation-aware attention weights across different relation spaces are used to sample multiple relation-specific subgraphs. Among these candidates, the two most influential subgraphs are selected as the two views for contrastive learning. For each node, its representations under these two relation-subgraph views form the positive pair, while representations of other nodes are treated as negatives, serving as a denoising regularizer. Candidate subgraphs are generated by combining relations with high attention weights, the sampling distribution is defined based on these weights, and the criterion for selecting the top-2 relation-specific variants is applied at the node level. GAT++ selects contextually relevant nodes within relation-specific semantic spaces, achieving the optimal performance.
>
> (a) In the current version of the paper, relation variants at each propagation layer are defined formally as subgraph variants across different relational spaces. GAT++ first samples multiple subgraphs and then selects the top two relation-specific variants to serve as the two contrastive views.
>
> (b) For the contrastive loss in Eq. (5), negatives are written as other nodes across the graph, i.e., $(o' \in G, o' \neq o)$, and the negative pairs are drawn from other nodes across the graph. $\tau$ is the temperature hyperparameter in the InfoNCE-style loss, which is determined by the specific scenario.
>
> The denoising encoder is positioned before the graph attention network and is responsible for refining the initial embeddings based on the raw interactions. Its output is propagated through subsequent GAT layers and multi-task training, enabling cleaner and more task-aligned feature representations for the downstream relation-aware attention and contrastive learning.
>
> W3, Q3 & Q5. We select representative methods within the recommendation scenario for comparison, and we evaluated all baselines using unified benchmarks with three runs. The testing results are consistent with existing work. Furthermore, to comprehensively evaluate the generalization capability of GAT++, we design a series of experiments targeting both controlled and real-world scenarios in the supplementary. Specifically, we investigate whether GAT++ can effectively identify informative relation paths, maintain strong performance in sparse or low-relational environments, and scale robustly to complex multi-relational graphs. We have proven that GAT++'s ability to reason across varying levels of relational complexity and semantic noise extends not only to recommendation tasks.
>
> W4. Our denoising encoder, the data generation module, is one of our key innovations. Among existing data generation methods, DR4SR+ is the most effective method.
>
> W5. We have fixed typos carefully.
>
> W7. In the Ablation Study (RQ3) section, we conduct a detailed analysis of how various ablation components, including the adaptive attention mechanism, contribute to the performance of GAT++.
>
> W8. For contrastive regularization and mutual information maximization, we have included detailed citations in the paper. The concept of robustness to high-order noise is also widely used in recommendation scenarios, and likewise, we have provided extensive references in the introduction and related work.
>
> Q4. Knowledge graph triples and user–item interaction graphs are two distinct types of knowledge graphs, so there is no risk of data leakage between them.

---

> > ### Comment · Reviewer_FaRs · 2025-11-20
> >
> > I appreciate the additional clarifications provided by the authors. After reviewing the rebuttal carefully, I find that some points are better explained now, but several key concerns remain unresolved.
> >
> > **Regarding W1:**
> >
> > The authors restate the three main modules and claim that GAT++ is the first to decouple attention weight distributions by relation. I still do not find this convincing. Prior work such as CompGCN, KGIN, and KGAT already employs relation specific parameterizations and heterogeneous attention. Rephrasing these mechanisms as a new form of decoupling does not establish meaningful novelty. The response does not provide evidence that the proposed idea is fundamentally different from the existing literature. My concern about limited novelty therefore remains.
> >
> > **Regarding W2, Q1, and Q2:**
> >
> > The additional explanation of multiplicative projection matrices and relation based subgraph sampling is helpful, but the method still lacks enough detail. The process for constructing and sampling relation specific subgraphs remains vague, and the choice of selecting the top two views is not supported by any empirical study or theoretical argument. I also do not see a formal description of the size of the candidate pool, the probability distribution for sampling, or the number of relations that participate at each propagation layer. What is missing is a precise mathematical formulation that clearly defines each step of the sampling and attention mechanism. Since the paper currently describes these procedures mainly at a conceptual level, the core operations of the model cannot be implemented or verified with confidence.
> >
> > Regarding W3, Q3, and Q5
> > The authors state that all baselines were run three times, and they refer to additional experiments in the supplementary material. However, the main paper still does not provide standard deviations, runtime measurements, memory consumption, or any discussion of training efficiency on a large dataset such as Dianping-Food. Without these elements, the very large reported improvements remain difficult to trust. My concerns about empirical transparency remain largely unaddressed.
> >
> > **Regarding W4:**
> >
> > My question is unaddressed.
> >
> > **Regarding W5:**
> >
> > The authors have corrected typos. I consider this resolved.
> >
> > **Regarding W7:**
> >
> > The authors point to the ablation study, which focuses on performance changes. However, the concern was about interpretability and insight into why the model works. There is still no qualitative analysis of the learned attention weights or relation distributions. This part remains only partially addressed.
> >
> > **Regarding W8:**
> >
> > The authors refer to citations about mutual information and robustness to high order noise. However, the manuscript still contains no theoretical justification, derivation, or empirical measurement related to mutual information. Citing related work does not replace providing actual evidence within this paper. The concern remains unresolved.
> >
> > **Regarding Q4:**
> >
> > The explanation that KG triples and user item interactions are separate graph types is reasonable. I consider this point resolved.
> >
> > **Overall assessment after rebuttal:**
> > Some minor clarity issues have been addressed, but major concerns about novelty, methodological precision, empirical rigor, and justification of design choices remain. The additional explanations are mostly descriptive and do not resolve the core issues that affect the validity and significance of the work. My scores therefore remain unchanged, and my recommendation stays the same.

---

> > > ### Author Response · Authors · 2025-11-22
> > > **Novelty of GAT++**
> > >
> > > We introduce GAT++, a graph attention framework that models multiple semantic relation spaces through relation-specific attention and saliency-aware aggregation, improving robustness to high-order noise in knowledge graphs. To our knowledge, GAT++ is the first to decouple attention weight distributions by relation. First, we propose a contrastive denoising regularizer that addresses semantic noise from high-order knowledge graph propagation by generating subgraph variants from the most influential relation spaces to enhance representation consistency and suppress irrelevant knowledge paths. Second, we propose a task denoising encoder that refines user-item interactions end-to-end in a task-specific manner. By eliminating the need for external data augmentation modules, this component enhances robustness and generalization under sparse supervision. Third, we conduct extensive experiments on multiple public benchmarks, where GAT++ consistently outperforms state-of-the-art recommendation models in accuracy and robustness across diverse datasets.

---

### Official Review · Reviewer_EJpo · 2025-11-01

**Soundness:** 2
**Presentation:** 2
**Contribution:** 2
**Rating:** 2
**Confidence:** 5

**Summary:**

The paper proposes GAT++, a relation-aware graph attention framework for knowledge-enhanced recommendation. It projects entities into relation-specific subspaces with adaptive attention, adds a contrastive denoising regularizer built from salient relation-subgraph variants, and introduces a personalized denoising encoder trained end-to-end. Experiments on Last.FM, Book-Crossing, and Dianping-Food report consistent gains, including a cold-start setting.

**Strengths:**

The encoder is described with a Transformer formulation, eliminating external generators and aligning with task objectives.

**Weaknesses:**

Mathematical/notation consistency issues: Mixed “Lnoize” (Eq. (5)) vs “LNoise” (Fig. 2), and ambiguous indexing in Eq. (4);

Outdated baseline coverage (2019–2020 only): As reported gains are only established over pre-2021 baselines, the experimental section currently provides limited evidence for contemporary competitiveness; stronger conclusions would require including recent SOTA baselines or justifying their omission.

**Questions:**

Could you precisely specify how the “relation-subgraph views” are constructed and sampled? In particular, (a) how are “relation variants” defined per layer and what saliency score selects the “top two” variants; (b) what is the negative-sample distribution (in-batch vs. memory queue) and the value/sensitivity range of the temperature τ

---

> ### Author Response · Authors · 2025-11-18
> **Response to Reviewer EJpo**
>
> W1. We have fixed typos carefully.
>
> W2. We use the recommendation scenario as the primary evaluation setting to demonstrate the effectiveness of our method. Furthermore, to comprehensively evaluate the generalization capability of GAT++, we design a series of experiments targeting both controlled and real-world scenarios in the supplementary. Specifically, we investigate whether GAT++ can effectively identify informative relation paths, maintain strong performance in sparse or low-relational environments, and scale robustly to complex multi-relational graphs. We have proven that GAT++'s ability to reason across varying levels of relational complexity and semantic noise extends not only to recommendation tasks.
>
> Q. The paper constructs and samples relation-subgraph views as follows: at each propagation layer, relation-aware attention weights across different relation spaces are used to sample multiple relation-specific subgraphs. Among these candidates, the two most influential subgraphs are selected as the two views for contrastive learning. For each node, its representations under these two relation-subgraph views form the positive pair, while representations of other nodes are treated as negatives, serving as a denoising regularizer. Candidate subgraphs are generated by combining relations with high attention weights, the sampling distribution is defined based on these weights, and the criterion for selecting the top-2 relation-specific variants is applied at the node level.
>
> (a) In the current version of the paper, relation variants at each propagation layer are defined formally as subgraph variants across different relational spaces. GAT++ first samples multiple subgraphs and then selects the top two relation-specific variants to serve as the two contrastive views.
>
> (b) For the contrastive loss in Eq. (5), negatives are written as other nodes across the graph, i.e., $(o' \in G, o' \neq o)$, and the negative pairs are drawn from other nodes across the graph. $\tau$ is the temperature hyperparameter in the InfoNCE-style loss, which is determined by the specific scenario.

---

### Official Review · Reviewer_kmrg · 2025-11-03

**Soundness:** 2
**Presentation:** 3
**Contribution:** 1
**Rating:** 4
**Confidence:** 5

**Summary:**

This paper proposes GAT++, a graph attention network for knowledge graph-enhanced recommendation systems that addresses semantic noise during multi-hop propagation. The method features three components: (1) adaptive relation-aware attention with relation-specific projections, (2) contrastive denoising regularization using multi-relation subgraphs, and (3) a personalized denoising encoder. Experiments on three datasets show up to 34.81% improvement in Recall@N over baselines.

**Strengths:**

1. The paper introduces a novel relation-aware attention mechanism that explicitly models multiple semantic relation spaces through learnable projection matrices, enabling fine-grained discrimination among heterogeneous relational dependencies in knowledge graphs.

2. Extensive experiments have been conducted across three diverse datasets from different domains (music, literature, and food), with comprehensive comparisons against multiple state-of-the-art baseline methods and thorough ablation studies demonstrating statistical significance.

3. The effectiveness of individual components has been well validated through systematic ablation studies, showing that each proposed module (adaptive attention, contrastive denoising, and personalized encoder) contributes meaningfully to the overall performance improvements.

**Weaknesses:**

1. The paper's title, abstract, and introduction fail to clearly specify that this is recommendation system research, creating significant confusion for readers. While the methodology section and experiments clearly focus on user-item recommendation tasks, the early sections present the work as general graph neural network research, which is misleading given the task-specific nature of the proposed solutions.

2. The paper proposes modifications to GAT architecture by introducing relation-specific projections and multi-space attention mechanisms. However, GAT has been extensively studied for many years with numerous architectural variants proposed. It is unclear how this paper makes a significant contribution to the already rich literature of GAT architecture designs, particularly given that the core innovation appears to be relatively incremental adaptations for recommendation scenarios.

**Questions:**

1. Why doesn't the paper clearly identify itself as recommendation system research in the title and early sections?

2. What are the fundamental technical contributions beyond existing GAT architectural variants that justify publication in a top-tier venue?

---

> ### Author Response · Authors · 2025-11-18
> **Response to Reviewer kmrg**
>
> W1 & Q1. We use the recommendation scenario as the primary evaluation setting to demonstrate the effectiveness of our method. Furthermore, to comprehensively evaluate the generalization capability of GAT++, we design a series of experiments targeting both controlled and real-world scenarios in the supplementary. Specifically, we investigate whether GAT++ can effectively identify informative relation paths, maintain strong performance in sparse or low-relational environments, and scale robustly to complex multi-relational graphs. We have proven that GAT++'s ability to reason across varying levels of relational complexity and semantic noise extends not only to recommendation tasks.
>
> W2 & Q2. We introduce GAT++, a graph attention framework that models multiple semantic relation spaces through relation-specific attention and saliency-aware aggregation, improving robustness to high-order noise in knowledge graphs. To our knowledge, GAT++ is the first to decouple attention weight distributions by relation. First, we propose a contrastive denoising regularizer that addresses semantic noise from high-order knowledge graph propagation by generating subgraph variants from the most influential relation spaces to enhance representation consistency and suppress irrelevant knowledge paths. Second, we propose a personalized denoising encoder that refines user-item interactions end-to-end in a task-specific manner. By eliminating the need for external data augmentation modules, this component enhances robustness and generalization under sparse supervision. Third, we conduct extensive experiments on multiple public benchmarks, where GAT++ consistently outperforms state-of-the-art recommendation models in accuracy and robustness across diverse datasets.

---

### Author Response · Authors · 2025-11-22
**Novelty of GAT++**

We introduce GAT++, a graph attention framework that models multiple semantic relation spaces through relation-specific attention and saliency-aware aggregation, improving robustness to high-order noise in knowledge graphs. To our knowledge, GAT++ is the first to decouple attention weight distributions by relation. First, we propose a contrastive denoising regularizer that addresses semantic noise from high-order knowledge graph propagation by generating subgraph variants from the most influential relation spaces to enhance representation consistency and suppress irrelevant knowledge paths. Second, we propose a task denoising encoder that refines user-item interactions end-to-end in a task-specific manner. By eliminating the need for external data augmentation modules, this component enhances robustness and generalization under sparse supervision. Third, we conduct extensive experiments on multiple public benchmarks, where GAT++ consistently outperforms state-of-the-art recommendation models in accuracy and robustness across diverse datasets.

---

> ### Author Response · Authors · 2025-11-25
> **Performance Comparison with GAT**
>
> The performance comparison of GAT and GAT++ shows a statistically substantial improvement in GAT++ \( p < 0.01 \), with “RI” indicating the average relative improvement.
>
> |                      | Recall@20 | Recall@50 | RI          |
> |----------------------|-----------|-----------|-------------|
> | GAT        | 0.0153    | 0.0457    |             |
> | **GAT++ (Ours)**     | **0.1100** | **0.1500** | **424.24\%** |
>
> |                      | AUC    | F1     | RI         |
> |----------------------|--------|--------|------------|
> | GAT        | 0.7080 | 0.6341 |            |
> | **GAT++ (Ours)**     | **0.8086** | **0.7240** | **14.18\%** |
>
> As shown in the Table, GAT++ substantially outperforms GAT, with an average improvement of 424.24\% in Recall@N across multiple recommendation metrics.

---

### Meta-Review · Area_Chair_iS9s · 2026-01-07

**Summary:**

This paper proposes GAT++, a graph attention framework for knowledge graph enhanced recommendation. The method combines an adaptive relation aware attention mechanism that projects entities into relation specific subspaces and learns relation dependent saliency and aggregation, a contrastive denoising regularizer that forms augmented views from relation specific subgraph variants to suppress high order propagation noise, and a Transformer based personalized denoising encoder to refine user and item interaction sequences end to end.

Across reviews, there is consistent concern that the claimed innovations may be incremental relative to existing work on heterogeneous or relation aware GNN recommenders. In particular, reviewers were not convinced that the relation specific attention and saliency mechanism clearly distinguishes itself from closely related prior designs, requesting clearer positioning and evidence demonstrating novelty. Reviewers also highlighted insufficient methodological clarity hindering reproducibility: the relation view subgraph construction, saliency based selection procedure, and contrastive sampling methods are inadequately specified, and notation and equations were described as ambiguous or inconsistent. Empirically, reviewers requested stronger transparency and validation, including reporting means and standard deviations over multiple experimental runs and seeds, clearer specification of statistical testing procedures, and reporting runtime and memory efficiency metrics given the increased method complexity. Additionally, reviewers noted that the title and introduction could more explicitly frame the work as a knowledge graph enhanced recommendation method to reduce potential confusion about its scope.

After rebuttal, the key concerns remained were insufficient novelty and unclear presentation. Given these concerns, although the paper addresses an important problem of semantic noise in multi hop knowledge graph propagation and reports promising results, we expect a better version considering all the comments.

**Reviewer Concerns:**

see metareview

**Reviewer Scores:**

see metareview

---

### Decision · Program_Chairs · 2026-01-26

Reject